# Self-Supervised Learning of Event-Based Optical Flow with Spiking Neural Networks

**Jesse J. Hagenaars**\*
j.j.hagenaars@tudelft.nl

**Federico Paredes-Vallés**\*
f.paredesvalles@tudelft.nl

**Guido C. H. E. de Croon**
g.c.h.e.decroon@tudelft.nl

Micro Air Vehicle Laboratory
Delft University of Technology, The Netherlands

## Abstract

The field of neuromorphic computing promises extremely low-power and low-latency sensing and processing. Challenges in transferring learning algorithms from traditional artificial neural networks (ANNs) to spiking neural networks (SNNs) have so far prevented their application to large-scale, complex regression tasks. Furthermore, realizing a truly asynchronous and fully neuromorphic pipeline that maximally attains the abovementioned benefits involves rethinking the way in which this pipeline takes in and accumulates information. In the case of perception, spikes would be passed as-is and one-by-one between an event camera and an SNN, meaning all temporal integration of information must happen inside the network. In this article, we tackle these two problems. We focus on the complex task of learning to estimate optical flow from event-based camera inputs in a self-supervised manner, and modify the state-of-the-art ANN training pipeline to encode minimal temporal information in its inputs. Moreover, we reformulate the self-supervised loss function for event-based optical flow to improve its convexity. We perform experiments with various types of recurrent ANNs and SNNs using the proposed pipeline. Concerning SNNs, we investigate the effects of elements such as parameter initialization and optimization, surrogate gradient shape, and adaptive neuronal mechanisms. We find that initialization and surrogate gradient width play a crucial part in enabling learning with sparse inputs, while the inclusion of adaptivity and learnable neuronal parameters can improve performance. We show that the performance of the proposed ANNs and SNNs are on par with that of the current state-of-the-art ANNs trained in a self-supervised manner.

## 1 Introduction

Neuromorphic hardware promises highly energy-efficient and low-latency sensing and processing thanks to its sparse and asynchronous nature. Event cameras capture brightness changes at microsecond resolution [17], while neuromorphic processors have demonstrated orders of magnitude lower energy consumption and latency compared to von Neumann architectures [11, 29]. To realize the full potential of such neuromorphic pipelines, we have to move towards an event-based communication and processing paradigm, where single events are passed as-is between the event-based sensor/camera and the neuromorphic processor running a spiking neural network (SNN), without processing or

---

\*Equal contribution, with alphabetical ordering.

35th Conference on Neural Information Processing Systems (NeurIPS 2021).

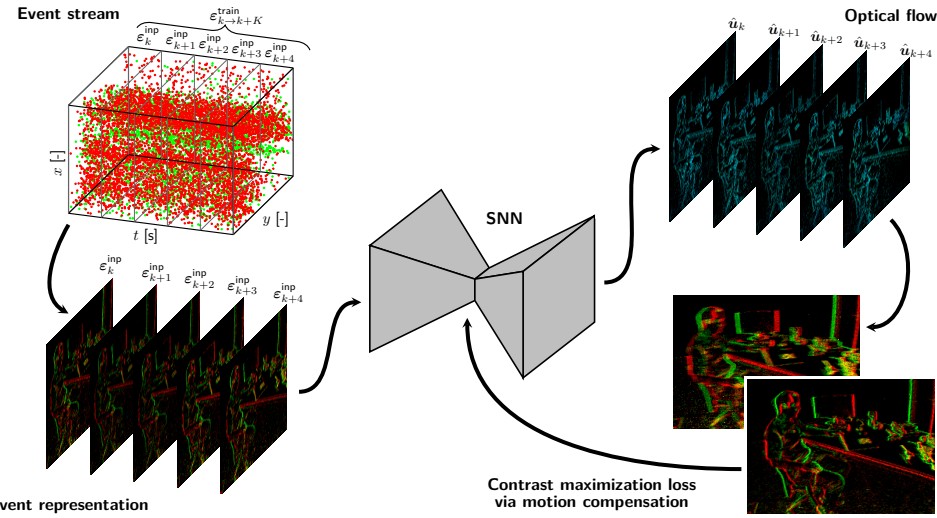

Figure 1: Self-supervised event-based optical flow pipeline for deep SNNs. In order of processing, the event stream is split into small partitions with the same number of events, which are formatted and then fed to the network in a sequential fashion. An optical flow map is predicted for each partition, associating every input event with a motion vector. Once a sufficient number of events has been processed, we perform a backward pass using our contrast maximization loss [16].

accumulation of any kind in between. Because of this, all temporal integration of information needs to happen inside the network itself. Most work on employing SNNs to event-based computer vision follows this approach [10, 14], but is limited to problems of limited temporal complexity (like classification). On the other hand, most state-of-the-art artificial neural network (ANN) pipelines for event-based computer vision combine a stateless feedforward architecture with encoding temporal information in the input [54, 56].

Apart from incompatible pipelines, one of the larger impediments to widespread neuromorphic adoption is the fact that learning algorithms designed for traditional ANNs do not transfer one-to-one to SNNs, which exhibit sparse, binary activity and more complex neuronal dynamics. On the one hand, this has driven research into the conversion of ANNs to SNNs without loss of accuracy, but with the promised efficiency gains [42]. On the other hand, it has limited the application of directly-trained SNNs in the computer vision domain to less complicated and often discrete problems like image classification [10, 14] on constrained datasets such as N-MNIST [33] or DVS128 Gesture [2].

Still, many ongoing developments in the area of direct SNN training are promising and may form building blocks for tackling more complex tasks. Surrogate gradients [32, 41, 45, 51], which act as stand-in for the non-differentiable spiking function in the backward pass, enable traditional backpropagation with few adjustments. Similarly, the inclusion of parameters governing the neurons' internal dynamics in the optimization was demonstrated to be beneficial [14, 38]. Many works also include some form of activity regularization to keep neurons from excessive spiking [5, 51] or to balance excitability through adaptation [4, 36]. Kickstarting initial activity (hence gradient flow) is often not the goal of these regularization terms, even though [51] shows that there is a narrow activity band in which learning is optimal. This ties in with the initialization of parameters, which has not been rigorously covered for SNNs with sparse inputs yet, leaving room for improvement.

Our goal is to demonstrate the potential of neuromorphic sensing and processing on a complex task. To this end, we tackle a real-world large-scale problem by learning, in a self-supervised fashion and using SNNs, to estimate the optical flow encoded in a continuous stream of events; a task that is usually tackled with deep, fully convolutional ANNs [54, 56]. By focusing on such a problem, we aim to identify and tackle emerging knowledge gaps regarding SNN training, while approximating a truly asynchronous pipeline.

In summary, the main contribution of this article is two-fold. First, we propose a novel self-supervised learning (SSL) framework for event-based optical flow estimation that puts emphasis on the networks' capacity to integrate temporal information from small, subsequent slices of events. This training

pipeline, illustrated in Fig. 1, is built around a reformulation of the self-supervised loss function from [56] that improves its convexity. Second, through this framework, we train the first set of deep SNNs that successfully solve the problem at hand. We validate our proposals through extensive quantitative and qualitative evaluations on multiple datasets[1]. Additionally, for the SNNs, we investigate the effects of elements such as parameter initialization and optimization, surrogate gradient shape, and adaptive neuronal mechanisms.

## 2  Related Work

Due to the potential of event cameras to enable low-latency optical flow estimation, extensive research has been conducted on this topic since these sensors were introduced [1, 6, 8, 15]. Regarding learning-based approaches, in [54], Zhu *et al.* proposed the first convolutional ANN for this task, which was trained in an SSL fashion with the supervisory signal coming from the photometric error between subsequent grayscale frames captured with the active pixel sensor (APS) of the DAVIS240C [7]. Alongside this network, the authors released the Multi-Vehicle Stereo Event Camera (MVSEC) dataset [55], the first event camera dataset with ground-truth optical flow estimated from depth and ego-motion sensors. A similar SSL approach was introduced in [48], but here optical flow was obtained through an ANN estimating depth and camera pose. Later, Zhu *et al.* refined their pipeline and, in [56], proposed an SSL framework around the contrast maximization for motion compensation idea from [15, 16]; with which, as explained in Section 3.2, the supervisory signal comes directly from the events and there is no need for additional sensors. More recently, Stoffregen and Scheerlinck *et al.* showed in [44] that, if trained with synthetic event sequences (from an event camera simulator [39]) and ground-truth data in a pure supervised fashion, the ANN from [54, 56] reaches higher accuracy levels when evaluated on MVSEC. Lastly, to hold up to the promise of high-speed optical flow, there has been a significant effort toward the miniaturization of optical flow ANNs [28, 35].

With respect to learning-based SNNs for optical flow estimation, only the works of Paredes-Vallés *et al.* [36] and Lee *et al.* [26, 27] are to be highlighted. In [36], the authors presented the first convolutional SNN in which motion selectivity emerges in an unsupervised fashion through Hebbian learning [22] and thanks to synaptic connections with multiple delays. However, this learning method limits the deployability of this architecture to event sequences with similar statistics to those used during training. On the other hand, in [26], the authors proposed a hybrid network, integrating spiking neurons in the encoder with ANN layers in the decoder, trained through the SSL pipeline from [54]. This architecture was later expanded in [27] with a secondary ANN-based encoder used to retrieve information from the APS frames. Lastly, SNNs have also been implemented in neuromorphic hardware for optical flow estimation [19], although this did not involve learning. Hence, until now, no one has yet attempted the SSL of optical flow with a pure SNN approach.

Most of the SNN work in other computer vision domains has so far been focused on discrete problems like classification [10, 14, 46, 53] and binary motion-based segmentation [34]. A notable exception is the work from Gehrig *et al.* [18], who propose a convolutional spiking encoder to continuously predict angular velocities from event data. However, until now, no one has yet attempted a dense (i.e., with per-pixel estimates) regression problem with deep SNNs that requires recurrency.

## 3  Method

### 3.1  Input event representation

An event camera consists of a pixel array that responds, in a sparse and asynchronous fashion, to changes in brightness through streams of events [17]. For an ideal camera, an event $e_i = (x_i, t_i, p_i)$ of polarity $p_i \in \{+, -\}$ is triggered at pixel $x_i = (x_i, y_i)^T$ and time $t_i$ whenever the brightness change since the last event at that pixel reaches the contrast sensitivity threshold for that polarity.

The great majority of learning-based models proposed to date for the problem of event-based optical flow estimation encode, in one form or another, spatiotemporal information into the input event representation before passing it to the neural architectures. This allows stateless (i.e., non-recurrent) ANNs to accurately estimate optical flow at the cost of having to accumulate events over relatively long time windows for their apparent motion to be perceivable. The most commonly used representations

---

[1]The project's code and additional material can be found at https://mavlab.tudelft.nl/event_flow/.

make use of multiple discretized frames of event counts [26, 27, 35, 44, 56] and/or the per-pixel average or the most recent event timestamps [28, 48, 54].

Ideally, SNNs would immediately receive spikes at event locations, which implies that temporal information should not be encoded in the input representation, but should be extracted by the network. To enforce this, we use a representation consisting only of per-pixel and per-polarity event counts, as in Fig. 1. This representation gets populated with consecutive, non-overlapping partitions of the event stream $\varepsilon_k^{\text{inp}} \doteq \{e_i\}_{i=0}^{N-1}$ (referred to as input partition) each containing a fixed number of events, $N$.

## 3.2 Self-supervised learning of optical flow via contrast maximization

We use the contrast maximization proxy loss for motion compensation [16] to learn to estimate optical flow from the continuous event stream in a self-supervised fashion. The idea behind this optimization framework is that accurate optical flow information is encoded in the spatiotemporal misalignments among the events triggered by the same portion of a moving edge (i.e., blur) and that, to retrieve it, one has to compensate for this motion (i.e., deblur the event partition). Knowing the per-pixel optical flow $\boldsymbol{u}(\boldsymbol{x}) = (u(\boldsymbol{x}), v(\boldsymbol{x}))^T$, the events can be propagated to a reference time $t_{\text{ref}}$ through:

$$\boldsymbol{x}_i' = \boldsymbol{x}_i + (t_{\text{ref}} - t_i)\boldsymbol{u}(\boldsymbol{x}_i) \tag{1}$$

In this work, we reformulate the deblurring quality measure proposed by Mitrokhin *et al.* [30] and Zhu *et al.* [56]: the per-pixel and per-polarity average timestamp of the image of warped events (IWE). The lower this metric, the better the event deblurring and the more accurate the optical flow estimation. We generate an image of the average timestamp at each pixel for each polarity $p'$ via bilinear interpolation:

$$T_{p'}(\boldsymbol{x};\boldsymbol{u}|t_{\text{ref}}) = \frac{\sum_j \kappa(x - x_j')\kappa(y - y_j')t_j}{\sum_j \kappa(x - x_j')\kappa(y - y_j') + \epsilon}$$
$$\kappa(a) = \max(0, 1 - |a|) \tag{2}$$
$$j = \{i \mid p_i = p'\}, \quad p' \in \{+, -\}, \quad \epsilon \approx 0$$

Previous works minimize the sum of the squared temporal images resulting from the warping process [35, 56]. However, we scale this sum prior to the minimization with the number of pixels with at least one warped event in order for the loss function to be convex:

$$\mathcal{L}_{\text{contrast}}(t_{\text{ref}}) = \frac{\sum_{\boldsymbol{x}} T_+(\boldsymbol{x};\boldsymbol{u}|t_{\text{ref}})^2 + T_-(\boldsymbol{x};\boldsymbol{u}|t_{\text{ref}})^2}{\sum_{\boldsymbol{x}} [n(\boldsymbol{x}') > 0] + \epsilon} \tag{3}$$

where $n(\boldsymbol{x}')$ denotes a per-pixel event count of the IWE. As shown in Appendix A, without the scaling, the loss function is not well-defined as the optimal solution is to always warp events with large timestamps out of the image space so they do not contribute to Eq. 2. Previous works circumvented this issue by limiting the maximum magnitude of the optical flow vectors that could be estimated through scaled TanH activations in the prediction layers [35, 56].

As in [56], we perform the warping process both in a forward ($t_{\text{ref}}^{\text{fw}}$) and in a backward fashion ($t_{\text{ref}}^{\text{bw}}$) to prevent temporal scaling issues during backpropagation. The total loss used to train our event-based optical flow networks is then given by:

$$\mathcal{L}_{\text{contrast}} = \mathcal{L}_{\text{contrast}}(t_{\text{ref}}^{\text{fw}}) + \mathcal{L}_{\text{contrast}}(t_{\text{ref}}^{\text{bw}}) \tag{4}$$
$$\mathcal{L}_{\text{flow}} = \mathcal{L}_{\text{contrast}} + \lambda \mathcal{L}_{\text{smooth}} \tag{5}$$

where $\lambda$ is a scalar balancing the effect of the two losses and $\mathcal{L}_{\text{smooth}}$ is a Charbonnier smoothness prior [9], as in [54, 56]. Since $\mathcal{L}_{\text{contrast}}$ does not propagate the error back to pixels without input events, we mask the output of our networks so that null optical flow vectors are returned at these pixel locations. Furthermore, we mask the computation of $\mathcal{L}_{\text{smooth}}$ so that this regularization mechanism only considers optical flow estimates from neighboring pixels with at least one event.

As hinted by the motion model in Eq. 1 and discussed in [16, 43], there has to be enough linear blur in the input event partition for $\mathcal{L}_{\text{contrast}}$ to be a robust supervisory signal. This is usually not the case in our training pipeline due to the small number of input events $N$ that we pass to the networks at each forward pass. For this reason, we define a secondary event partition, the so-called training partition

$\varepsilon_{k \to k+K}^{\text{train}} \doteq \{(\varepsilon_i^{\text{inp}}, \hat{\boldsymbol{u}}_i)\}_{i=k}^K$, which is a buffer that gets populated every forward pass with an input event partition and its corresponding optical flow estimates. At training time, we perform a backward pass with the content of the buffer using backpropagation through time once it contains $K$ successive event-flow tuples, after which we detach the state of the networks from the computational graph and clear the buffer. Note that $\mathcal{L}_{\text{smooth}}$ is also applied in the temporal dimension by smoothing optical flow estimates at the same pixel location from adjacent tuples.

### 3.3 Spiking neuron models

We compare various spiking neuron models from literature on the task of event-based optical flow estimation. All models are based on the leaky-integrate-and-fire (LIF) neuron, whose membrane potential $U$ and synaptic input current $I$ at timestep $k$ can be written as:

$$U_i^k = (1 - S_i^{k-1})\alpha U_i^{k-1} + (1 - \alpha)I_i^k \tag{6}$$

$$I_i^k = \sum_j W_{ij}^{\text{ff}} S_j^k + \sum_r W_{ir}^{\text{rec}} S_r^{k-1} \tag{7}$$

where $j$ and $r$ denote presynaptic neurons while $i$ is for postsynaptic, $\alpha$ is the membrane decay or leak, $S \in \{0, 1\}$ a neuron spike, and $W^{\text{ff}}$ and $W^{\text{rec}}$ feedforward and recurrent connections, respectively. Membrane decays can either be fixed or learned. A neuron fires an output spike $S$ if the membrane potential exceeds a threshold $\theta$, which can either be fixed, learned, or adaptive (see below). Firing also triggers a reset of $U$, which is either *hard*, as in Eq. 6, or *soft*, as in [4]. The former is said to be more suitable for deeper networks, as it gets rid of errors accumulated by the surrogate gradient [25].

Following [4], we introduce an adaptive threshold to make up the adaptive LIF (ALIF) model. A second state variable $T$ acts as a low-pass filter over the output spikes, adapting the firing threshold based on the neuron's activity:

$$\theta_i^k = \beta_0 + \beta_1 T_i^k \tag{8}$$

$$T_i^k = \eta T_i^{k-1} + (1 - \eta)S_i^{k-1} \tag{9}$$

where $\beta_{\{0,1\}}$ are (learnable) constants, and $\eta$ is the (learnable) threshold decay/leak. ALIF's equations for $U$ and $I$ are identical to the LIF formulation. By decaying the threshold very slowly, $T$ can act as a longer-term memory of the neuron [5].

Instead of postsynaptic adaptivity, we can keep a trace of presynaptic activity and use that to regularize neuron firing, giving the presynaptic LIF (PLIF) model. The authors of [36] implement this kind of adaptation mechanism by subtracting a presynaptic trace $P$ from the input current:

$$I_i^k = \sum_j W_{ij}^{\text{ff}} S_j^k + \sum_r W_{ir}^{\text{rec}} S_r^{k-1} - \rho_0 P_i^k \tag{10}$$

$$P_i^k = \rho_1 P_i^{k-1} + \frac{1 - \rho_1}{|R_i|} \sum_{j \in R_i} S_j^{k-1} \tag{11}$$

where $\rho_{\{0,1\}}$ are (learnable) addition and decay constants, and $R_i$ is the set of receptive fields of neuron $i$ over all channels (i.e., the second term in Eq. 11 is an average pooling averaged over all channels). Adaptation based on presynaptic instead of postsynaptic activity minimizes adaptation delay, making it especially suited to the fast-changing nature of event data [36]. In this spirit, we also propose the XLIF model, a crossover between ALIF and PLIF, which adapts its threshold based on presynaptic activity:

$$\theta_i^k = \beta_0 + \beta_1 P_i^k \tag{12}$$

As surrogate gradient for the spiking function $\sigma$, we opt for the derivative of the inverse tangent $\sigma'(x) = \text{aTan}' = 1/(1 + \gamma x^2)$ [14] because it is computationally cheap, with $\gamma$ being the surrogate width and $x = U - \theta$. In order to ensure gradient flow (hence learning) in the absence of neuron firing, the width should be sufficient to cover at least a range of subthreshold membrane potentials, while the height should be properly scaled (i.e., $\leq 1$) for stable learning [51]. Exact shape is of less importance for final accuracy. Further details on all hyperparameters can be found in Appendix G.

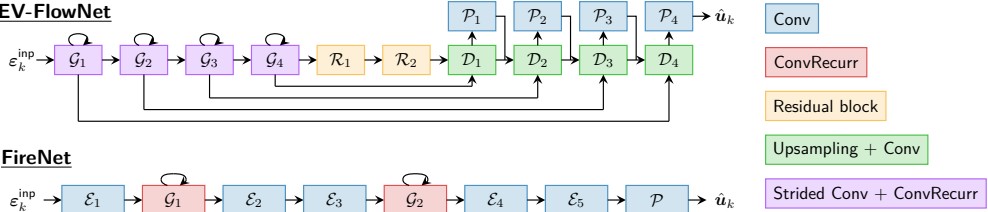

Figure 2: Schematic of the base neural networks used in this work. All evaluated variants inherit from these architectures and only vary the neuron model and/or the convolutional recurrent layer.

## 3.4 Network architectures

We evaluate the two trends on neural network design for event cameras through (spiking) recurrent variants of EV-FlowNet [54] (encoder-decoder) and FireNet [40] (lightweight, no downsampling). An overview of the evaluated architectures can be found in Fig. 2. The use of explicit recurrent connections in all our ANNs and SNNs is justified through the ablation study in Appendix D.

The base architecture referred to as EV-FlowNet is a recurrent version of the network proposed in [54]. Once represented as in Section 3.1, the input event partition is passed through four recurrent encoders performing strided convolution followed by ConvGRU [3] with output channels doubling after each encoder (starting from 32), two residual blocks [13, 20], and four decoder layers that perform bilinear upsampling followed by convolution. After each decoder, there is a (concatenated) skip connection from the corresponding encoder, as well as a depthwise (i.e., $1 \times 1$) convolution to produce a lower scale flow estimate, which is then concatenated with the activations of the previous decoder. The $\mathcal{L}_{\text{flow}}$ loss (see Eq. 5) is applied to each intermediate optical flow estimate via upsampling. All layers use $3 \times 3$ kernels and ReLU activations except for the prediction layers, which use TanH activations.

The FireNet architecture in Fig. 2 is an adaptation of the lightweight network proposed in [40], which was originally designed for event-based image reconstruction. However, as shown in [35], this architecture is also suitable for fast optical flow estimation. The base architecture consists of five encoder layers that perform single-strided convolution, two ConvGRUs, and a final prediction layer that performs depthwise convolution. All layers have 32 output channels and use $3 \times 3$ kernels and ReLU activations except for the final layer, which uses a TanH activation.

Based on these architectures, we have designed several variants: (i) RNN-EV-FlowNet and RNN-FireNet, which use vanilla ConvRNNs (see Appendix B) instead of ConvGRUs; (ii) Leaky-EV-FlowNet and Leaky-FireNet, which use ConvRNNs and whose neurons are stateful cells with leaks (for a more direct comparison with the SNNs, see Appendix B); and (iii) SNN-EV-FlowNet and SNN-FireNet (with SNN being LIF, ALIF, PLIF or XLIF), the SNN variants that use ConvRNNs and whose neurons are spiking and stateful according to the neuron models in Section 3.3.

The prediction layers of all SNN variants are kept real-valued with TanH activation, acting as a learned decoder from binary spikes to a dense optical flow estimate. The first layer of the SNNs can likewise be viewed as a learned spike encoder, receiving integer event counts and emitting spikes.

## 4 Experiments

To highlight the robustness of our SSL pipeline, we train our networks on the indoor forward-facing sequences from the UZH-FPV Drone Racing Dataset [12], which is characterized by a much wider distribution of optical flow vectors than the datasets that we use for evaluation, i.e., MVSEC [54], High Quality Frames (HQF) [44], and the Event-Camera Dataset (ECD) [31]. The selected training sequences consist of approximately 15 minutes of event data that we split into 140 $128 \times 128$ (randomly cropped) sequences with 500k events each. We further augment this data using random horizontal, vertical, and polarity flips.

Our framework is implemented in PyTorch. We use the Adam optimizer [24] and a learning rate of 0.0002, and train with a batch size of 8 for 100 epochs. We clip gradients based on a global norm of 100. We fix the number of events for each input partition to $N = 1$k, while we use 10k events for each training event partition. This is equivalent to $K = 10$ forward passes per backward pass (i.e.,

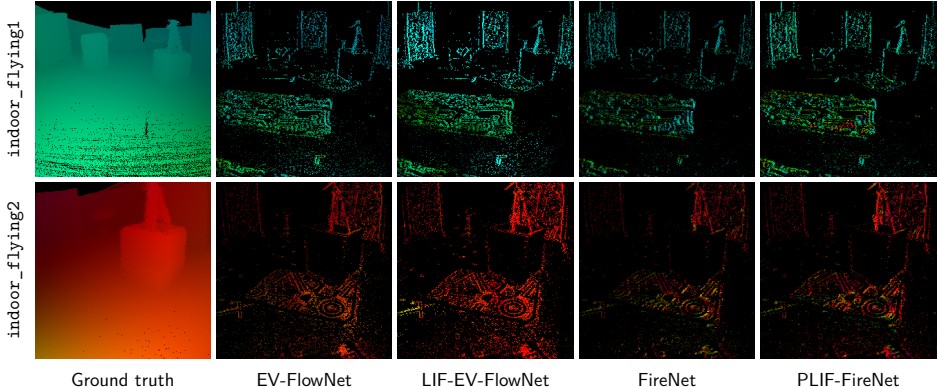

Figure 3: Qualitative evaluation of our best performing ANNs and SNNs on sequences from the MVSEC dataset [55]. The optical flow color-coding scheme can be found in Appendix C.

the network's unrolling), as described in Section 3.2 and illustrated in Fig. 1. Lastly, we empirically set the scaling weight for $\mathcal{L}_{\text{smooth}}$ to $\lambda = 0.001$.

We evaluated our architectures on the MVSEC dataset [55] with the ground-truth optical flow data provided by Zhu *et al.* in [54], which was generated at each APS frame timestamp, and scaled to be the displacement for the duration of one (dt $= 1$) and four (dt $= 4$) APS frames. Optical flow predictions were also generated at each frame timestamp by using all the events in the time window as input for dt $= 1$, or $25\%$ of the window events at a time for dt $= 4$ (due to the larger displacements). For comparison against the ground truth, the predicted optical flow is converted from units of pixels/partition to units of pixel displacement by multiplying it with $\text{dt}_{\text{gt}}/\text{dt}_{\text{input}}$. We compare our recurrent ANNs and SNNs against the state-of-the-art on self-supervised event-based optical flow estimation: the original (non-recurrent) EV-FlowNet [54] trained with either photometric error as in [54] or contrast maximization [56], and the hybrid SNN-ANN network from [26]. Quantitative results of this evaluation are presented in Table 1. We report the average endpoint error (AEE) and the percentage of points with AEE greater than 3 pixels and $5\%$ of the magnitude of the optical flow vector, denoted by $\%_{\text{Outlier}}$, over pixels with valid ground-truth data and at least one input event. Qualitative results of our best performing networks on this dataset are shown in Fig. 3.

For the sake of completeness, as in [35, 44] we also evaluate our architectures on the ECD [31] and HQF [44] datasets. The details and results of this evaluation can be found in Appendix C. Due to the lack of ground-truth data in these datasets, we assess the quality of the estimated optical flow based on metrics derived from the contrast maximization framework [15, 16].

Regarding the SNN variants, the results in Table 1, Fig. 3 and Appendix C correspond to networks whose neuronal parameters (i.e., leaks, thresholds, adaptive mechanisms) were also optimized when applicable. See Appendix E for an ablation study on the learnable parameters; further details regarding parameter settings and initialization are given in Appendix G.

### 4.1 Evaluation of the ANN and SNN architectures

Firstly, the quantitative results in Table 1 confirm the validity of the proposed SSL framework for event-based optical flow estimation with recurrent networks. As shown, our base architectures EV-FlowNet and FireNet perform on par with the current state-of-the-art, even though these non-recurrent networks from literature encode explicit temporal information in their input event representations, and were trained on other very similar sequences from MVSEC [55] to prevent the input statistics from deviating from the training distribution during inference [26, 54, 56]. Since we train on a very different dataset [12], this on-par performance also confirm the generalizability of our ANNs and SNNs to distinctly different scenes and distributions of optical flow vectors. This claim is further supported by qualitative results in Fig. 3 and additional results in Appendix C.

Secondly, from the comparison between our base ANN architectures and their spiking counterparts without adaptation mechanisms (i.e., LIF-EV-FlowNet and LIF-FireNet), we can conclude that, although there is a general increase in the AEE and the percentage of outliers when going spiking,

Table 1: Quantitative evaluation on MVSEC [55]. For each sequence, we report the AEE (lower is better, ↓) in pixels and the percentage of outliers, $\%_{\text{Outlier}}$ (↓). Best in bold, runner up underlined.

| dt = 1 | outdoor_day1 | | indoor_flying1 | | indoor_flying2 | | indoor_flying3 | |
|---|---|---|---|---|---|---|---|---|
| | AEE | $\%_{\text{Outlier}}$ | AEE | $\%_{\text{Outlier}}$ | AEE | $\%_{\text{Outlier}}$ | AEE | $\%_{\text{Outlier}}$ |
| EV-FlowNet* [54] | 0.49 | 0.20 | 1.03 | 2.20 | 1.72 | 15.10 | 1.53 | 11.90 |
| EV-FlowNet* [56] | **0.32** | **0.00** | **0.58** | **0.00** | **1.02** | **4.00** | **0.87** | **3.00** |
| Hybrid-EV-FlowNet* [26] | 0.49 | - | 0.84 | - | 1.28 | - | 1.11 | - |
| EV-FlowNet | 0.47 | 0.25 | 0.60 | 0.51 | 1.17 | 8.06 | 0.93 | 5.64 |
| RNN-EV-FlowNet | 0.56 | 1.09 | 0.62 | 0.97 | 1.20 | 8.82 | 0.93 | 5.51 |
| Leaky-EV-FlowNet | 0.53 | 0.28 | 0.71 | 0.60 | 1.43 | 11.37 | 1.14 | 8.12 |
| LIF-EV-FlowNet | 0.53 | 0.33 | 0.71 | 1.41 | 1.44 | 12.75 | 1.16 | 9.11 |
| ALIF-EV-FlowNet | 0.57 | 0.42 | 1.00 | 2.46 | 1.78 | 17.69 | 1.55 | 15.24 |
| PLIF-EV-FlowNet | 0.60 | 0.52 | 0.75 | 0.85 | 1.52 | 13.38 | 1.23 | 9.48 |
| XLIF-EV-FlowNet | 0.45 | 0.16 | 0.73 | 0.92 | 1.45 | 12.18 | 1.17 | 8.35 |
| FireNet | 0.55 | 0.35 | 0.89 | 1.93 | 1.62 | 14.65 | 1.35 | 10.64 |
| RNN-FireNet | 0.62 | 0.52 | 0.96 | 2.60 | 1.77 | 17.55 | 1.48 | 13.60 |
| Leaky-FireNet | 0.52 | 0.41 | 0.90 | 2.66 | 1.67 | 16.09 | 1.43 | 13.16 |
| LIF-FireNet | 0.57 | 0.40 | 0.98 | 2.48 | 1.77 | 16.40 | 1.50 | 12.81 |
| ALIF-FireNet | 0.62 | 0.45 | 1.04 | 3.02 | 1.85 | 18.88 | 1.58 | 15.00 |
| PLIF-FireNet | 0.56 | 0.38 | 0.90 | 1.93 | 1.67 | 14.47 | 1.41 | 11.17 |
| XLIF-FireNet | 0.54 | 0.34 | 0.98 | 2.75 | 1.82 | 18.19 | 1.54 | 14.57 |

| dt = 4 | | | | | | | | |
|---|---|---|---|---|---|---|---|---|
| EV-FlowNet* [54] | 1.23 | **7.30** | 2.25 | 24.70 | 4.05 | 45.30 | 3.45 | 39.70 |
| EV-FlowNet* [56] | 1.30 | 9.70 | 2.18 | 24.20 | 3.85 | 46.80 | 3.18 | 47.80 |
| Hybrid-EV-FlowNet* [26] | **1.09** | - | 2.24 | - | **3.83** | - | 3.18 | - |
| EV-FlowNet | 1.69 | 12.50 | **2.16** | **21.51** | 3.90 | **40.72** | **3.00** | **29.60** |
| RNN-EV-FlowNet | 1.91 | 16.39 | 2.23 | 22.10 | 4.01 | 41.74 | 3.07 | 30.87 |
| Leaky-EV-FlowNet | 1.99 | 17.86 | 2.59 | 30.71 | 4.94 | 54.74 | 3.84 | 42.33 |
| LIF-EV-FlowNet | 2.02 | 18.91 | 2.63 | 29.55 | 4.93 | 51.10 | 3.88 | 41.49 |
| ALIF-EV-FlowNet | 2.13 | 20.96 | 3.81 | 50.36 | 6.40 | 66.03 | 5.53 | 61.07 |
| PLIF-EV-FlowNet | 2.24 | 23.76 | 2.80 | 34.34 | 5.21 | 52.98 | 4.12 | 45.31 |
| XLIF-EV-FlowNet | 1.67 | 12.69 | 2.72 | 31.69 | 4.93 | 51.36 | 3.91 | 42.52 |
| FireNet | 2.04 | 20.93 | 3.35 | 42.50 | 5.71 | 61.03 | 4.68 | 53.42 |
| RNN-FireNet | 2.35 | 24.31 | 3.64 | 46.54 | 6.33 | 63.89 | 5.20 | 56.60 |
| Leaky-FireNet | 1.96 | 18.26 | 3.42 | 42.03 | 5.92 | 58.80 | 4.98 | 52.57 |
| LIF-FireNet | 2.12 | 21.00 | 3.72 | 48.27 | 6.27 | 64.16 | 5.23 | 58.43 |
| ALIF-FireNet | 2.36 | 25.82 | 3.94 | 52.35 | 6.65 | 67.61 | 5.60 | 61.93 |
| PLIF-FireNet | 2.11 | 20.64 | 3.44 | 44.02 | 5.94 | 64.02 | 4.98 | 57.53 |
| XLIF-FireNet | 2.07 | 18.83 | 3.73 | 47.89 | 6.51 | 67.25 | 5.43 | 60.59 |

*Non-recurrent ANNs with input event representations encoding spatiotemporal information, as described in [26, 54, 56].

the proposed SNNs are still able to produce high quality event-based optical flow estimates. In fact, according to Table 1, the main drop in accuracy does not come from the incorporation of the spiking function (and the selection of aTan′ as surrogate gradient), but mainly from the use of vanilla convolutional recurrent layers instead of gated recurrent units. As shown, our spiking LIF architectures perform very close to their RNN and leaky counterparts, despite the latter being ANNs. This highlights the important need for more powerful convolutional recurrent units for SNNs, similar to ConvLSTMs [47] and ConvGRUs [3] for ANNs, as this would narrow the performance gap between these two processing modalities according to our observations. Interestingly, a previous comparison of the performance of recurrent ANNs and SNNs for event-based classification [21] suggested similar improvements to SNN units.

## 4.2 Impact of adaptive mechanisms for spiking neurons

Table 1 and Appendix C also allow us to draw conclusions about the effectiveness of the adaptive mechanisms for spiking neurons introduced in Section 3.3. For both EV-FlowNet and FireNet, we observe that threshold adaptation based on postsynaptic activity (i.e., the ALIF model) performs

worse compared to other models. The loss curves in Appendix F support this observation. While the ALIF model was shown to be effective for learning long temporal dependencies from relatively low-dimensional data as in [4, 5, 49], the adaptation delay introduced by relying on a postsynaptic signal seems detrimental when working with fast-changing, high-dimensional event data. This is in line with suggestions by Paredes-Vallés *et al.* in [36], who use presynaptic adaptation for this reason. Our own results with presynaptic adaptation (i.e., PLIF and XLIF models) are somewhat inconclusive. While PLIF performs better in the case of FireNet, this is not the case for EV-FlowNet. On the other hand, XLIF's performance is very similar to the LIF model for both FireNet and EV-FlowNet architectures. Based on these observations, we think that adaptivity based on presynaptic activity should be considered for further development. In this regard, the XLIF model has the advantage that it is able to generate activity (leading to gradient flow, and thus learning) even for very small inputs, whereas PLIF is incapable of this for a given threshold (because $P$ is always positive). A more detailed comparison of activity levels for the different variants is given in Appendix I, along with an approximation of the energy efficiency gains of SNNs compared to ANNs.

### 4.3 Further lessons on training deep SNNs

Multiple problems arise when training deep SNNs for a regression task that involves sparse data, as is done here. Regarding learning, we find that gradient vanishing poses the main issue. Even considering dense inputs/loss and a shallow (in timesteps or in layers) SNN, sufficient gradient flow is a result of wide enough (in our case, covering at least $|U - \theta| \leq \theta$) and properly scaled (i.e., $\leq 1$) surrogate gradients [25, 49, 51], and parameter initializations that lead to non-negligible amounts of spiking activity [51]. Sparse data and deep networks make finding the proper settings more difficult, and for this reason, we have tried to increase the robustness of various of these hyperparameter settings. First, we looked at the learning performance and gradient flow of networks with various surrogate gradient shapes and widths. Compared to the aTan$'$ surrogate specified in Section 3.3, SuperSpike [50] with $\gamma = 10$ and $\gamma = 100$ (both narrower) show little learning due to negligible gradient flow (see Appendix H for more details). One way of reducing the effect of a too narrow surrogate gradient would be to trigger spiking activity through regularization terms in the loss function, as done in, e.g., [5, 51]. These form a direct connection between loss and the neuron in question, bypassing most of the gradient vanishing that would happen in later layers. We tried the variant proposed in [51], which is aimed at achieving at least a certain fraction of neurons to be active at any given time. With this fraction set to $5\%$, we saw that for SuperSpike with $\gamma = 10$ there was some learning happening, while for $\gamma = 100$ there was no effect. Plots of the loss curves and gradient magnitudes are available in Appendix H. Of course, more research into these and other regularization methods is necessary. Alternatively, as done in [25], batch normalization (or other presynaptic normalization mechanisms) could be used to ensure proper activity and gradient flow.

Regarding the network output, there seems to be an intuitive gap between classification and regression tasks, with the latter requiring a higher resolution to be solved successfully. In our view, there are two aspects to this that might pose an issue to SNNs. First, given the single prediction layer that the here-presented SNNs have to go from binary to real-valued activations, one could expect a loss in output resolution compared to equivalent ANN architectures. Second, even for moderate activity levels, the outputs of a spiking layer can be much larger in magnitude than an equivalent ANN layer, even for comparable parameter initializations. Intuitive solutions to these shortcomings are (i) to increase the number of channels to increase the resolution, and (ii) to initialize the weights of the non-spiking prediction layer as to have a smaller magnitude. While increasing the number of output channels in $\mathcal{E}_5$ (LIF-FireNet, see Fig. 2) did not lead to significantly improved performance or learning speed, decreasing the initialization magnitude of the weights in layer $\mathcal{P}$ did. As the loss curves in Appendix H show, the improved initialization leads to faster convergence and less variability across neuron models and the selection of learnable parameters.

## 5 Conclusion

In this article, we presented the first set of deep SNNs to successfully solve the real-world large-scale problem of event-based optical flow estimation. To achieve this, we first reformulated the state-of-the-art training pipeline for ANNs to considerably shorten the time windows presented to the networks, approximating the way in which SNNs would receive spikes directly from the event camera. Additionally, we reformulated the state-of-the-art self-supervised loss function to

improve its convexity. Prior to their training with this framework, we augmented several ANN architectures from literature with explicit and/or implicit recurrency, besides the addition of the spiking behavior. Extensive quantitative and qualitative evaluations were conducted on multiple datasets. Results confirm not only the validity of our training pipeline, but also the on-par performance of the proposed set of recurrent ANNs and SNNs with the self-supervised state-of-the-art. To the best of our knowledge, and especially due to the addition of explicit recurrent connections, the proposed SNNs correspond to the most complex spiking networks in the computer vision literature, architecturally speaking. For the SNNs, we also conducted several additional studies and (i) concluded that parameter initialization and the width of the surrogate gradient have a significant impact on learning: smaller weights in the prediction layer speed up convergence, while a too narrow surrogate gradient prevents learning altogether; and (ii) observed that adaptive mechanisms based on presynaptic activity outperform those based on postsynaptic activity, and perform similarly or better than the baseline without adaptation. Overall, we believe this article sets the groundwork for future research on neuromorphic processing for not only the event-based structure-from-motion problem, but also for other, similarly complex computer vision applications. For example, our results suggest the need for more powerful recurrent units for SNNs. On another note, future work should also focus on the implementation of these deep SNNs on neuromorphic hardware, as it is there where these architectures excel due to the power efficiency that their sparse and asynchronous nature brings.

## Acknowledgments and Disclosure of Funding

The authors would like to thank the anonymous reviewers for their constructive feedback and suggestions, and Kirk Y. W. Scheper for his insights on self-supervised learning of event-based optical flow. This work was supported by funding from NWO (grants NWA.1292.19.298 and 612.001.701).

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
