# A   Convexity of the self-supervised loss function

To evaluate the convexity of the self-supervised loss function for event-based optical flow estimation from [56] and the adaptation that we propose in this work, we conducted an experiment with two partitions of 40k events from the ECD dataset [31]. In this experiment, for the selected partitions, we computed the value of Eq. 4 (with and without the scaling) for four sets of optical flow vectors given by:

$$\boldsymbol{u}_s(d) = \{(u : u = g(i, d), v : v = g(j, d)), i \in \{0, 1, ..., 128\}, j \in \{0, 1, ..., 128\}\} \tag{13}$$

$$g(x, d) = \frac{2xd}{128} - d \tag{14}$$

where $d$ denotes the per-axis maximum displacement, which is drawn from the set $D = \{128, 256, 512, 1024\}$. This is equivalent to performing a grid search for the lowest $\mathcal{L}_{\text{contrast}}$ over an optical flow space ranging from $(-d, -d)$ to $(d, d)$ with 128 samples for each axis. Fig. 4 highlights the main difference between the original and our adapted formulation. Although for the smaller values of $d$ the two normalized losses look qualitatively similar, for larger values it is possible to discern that the original $\mathcal{L}_{\text{contrast}}$ is not convex, and that its optimal solution is to throw events out of the image space during the warping process so they do not contribute to the computation of the loss. On the contrary, the scaling that we propose in Section 3.2 fixes this issue, and results in a convex loss function for any value of $d$.

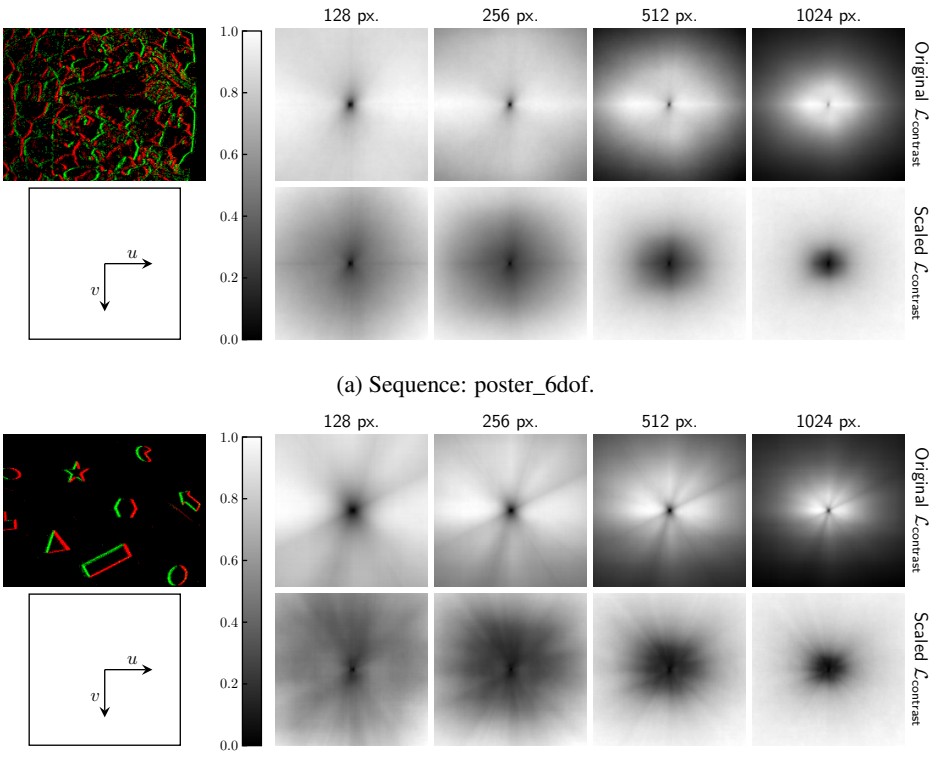

(a) Sequence: poster_6dof.

(b) Sequence: shapes_6dof.

Figure 4: Illustration of the effect of scaling $\mathcal{L}_{\text{contrast}}$ for different optical flow vectors for two event partitions from the Event-Camera Dataset [31]. Numbers on top indicate the maximum per-axis pixel displacement for each column.

# B   Clarifications on implicit and explicit recurrency

The definition of the implicit temporal dynamics of the leaky, non-spiking variants of our EV-FlowNet and FireNet base architectures closely resembles that of the membrane potential for spiking neurons

(see Eq. 6) but without the reset mechanism. With ReLU as non-linearity, the activation $Y$ of a leaky neuron is given by:

$$Y_i^k = \text{ReLU}\left(\alpha Y_i^{k-1} + (1-\alpha)\sum_j W_{ij}^{\text{ff}} I_i^k\right) \tag{15}$$

where $j$ and $i$ denote presynaptic and postsynaptic neurons respectively, $\alpha$ is the decay or leak of the neuron, $k$ the timestep, and $W^{\text{ff}}$ the feedforward weights multiplying the input signal $I$.

Regarding explicit recurrency, there is a slight difference between the vanilla ConvRNN layers used in our SNN and ANN architectures. On the one hand, the ConvRNNs that we use in our SNNs are defined through Eqs. 6 and 7 with two convolutional gates, one for the input and one for the recurrent signal, followed by the spiking function. On the other hand, the ConvRNNs in our ANNs are characterized by the same two convolutional gates but in this case followed by a TanH activation, and thereafter by a third output gate with ReLU activation. This augmentation was introduced to improve the convergence of the RNN and leaky variants of the base ANN architectures. From the results in Table 1 and Appendix C, we can observe that, despite this small difference, our SNNs perform on-par with their RNN and leaky counterparts.

## C   Self-supervised evaluation and additional qualitative results

Apart from the quantitative and qualitative evaluation on the MVSEC dataset [55] included in Section 4, we also evaluate our architectures on the ECD [31] and HQD [44] datasets, as in [35, 44]. Since these datasets lack ground-truth data, we use the Flow Warp Loss (FWL) [44], which measures the sharpness of the IWE relative to that of the original event partition using the variance as a measure of the contrast of the event images [16]. In addition to FWL, we propose the Ratio of the Squared Average Timestamps (RSAT) as a novel, alternative metric to measure the quality of the optical flow without ground-truth data. Contrary to FWL, RSAT makes use of Eq. 3 to measure the contrast of the event images and is defined as:

$$\text{RSAT} \doteq \frac{\mathcal{L}_{\text{contrast}}(t_{\text{ref}}^{\text{fw}}|\boldsymbol{u})}{\mathcal{L}_{\text{contrast}}(t_{\text{ref}}^{\text{fw}}|\boldsymbol{0})} \tag{16}$$

where $\text{RSAT} < 1$ implies that the predicted optical flow is better than a baseline consisting of null vectors. Since both FWL and RSAT are sensitive to the number of input events [36], we set $N = 15\text{k}$ events for all sequences in this evaluation. Quantitative results of this evaluation can be found in Table 2, while qualitative results on these datasets can are shown in Fig. 6. The optical flow color-coding scheme is given in Fig. 5.

Table 2: Quantitative evaluation on the ECD [31] and HQF [44] datasets. For each dataset, we report the mean FWL [44] (higher is better, ↑) and RSAT (↓). Best in bold, runner up underlined.

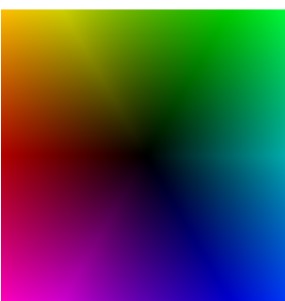

Figure 5: Optical flow color-coding scheme. Direction is encoded in color hue, and speed in color brightness.

| | ECD | | HQF | |
| --- | --- | --- | --- | --- |
| | FWL | RSAT | FWL | RSAT |
| EV-FlowNet | 1.31 | 0.94 | 1.37 | 0.92 |
| RNN-EV-FlowNet | 1.36 | 0.95 | 1.45 | 0.93 |
| Leaky-EV-FlowNet | 1.34 | 0.95 | 1.39 | 0.93 |
| LIF-EV-FlowNet | 1.21 | 0.95 | 1.24 | 0.94 |
| ALIF-EV-FlowNet | 1.17 | 0.98 | 1.21 | 0.98 |
| PLIF-EV-FlowNet | 1.24 | 0.95 | 1.28 | 0.93 |
| XLIF-EV-FlowNet | 1.23 | 0.95 | 1.25 | 0.93 |
| FireNet | 1.43 | 0.99 | 1.57 | 0.99 |
| RNN-FireNet | 1.34 | 0.99 | 1.42 | 0.99 |
| Leaky-FireNet | 1.40 | 0.99 | 1.52 | 0.99 |
| LIF-FireNet | 1.28 | 0.99 | 1.34 | 1.00 |
| ALIF-FireNet | 1.35 | 1.00 | 1.49 | 1.00 |
| PLIF-FireNet | 1.30 | 0.97 | 1.35 | 0.98 |
| XLIF-FireNet | 1.29 | 0.99 | 1.39 | 0.99 |

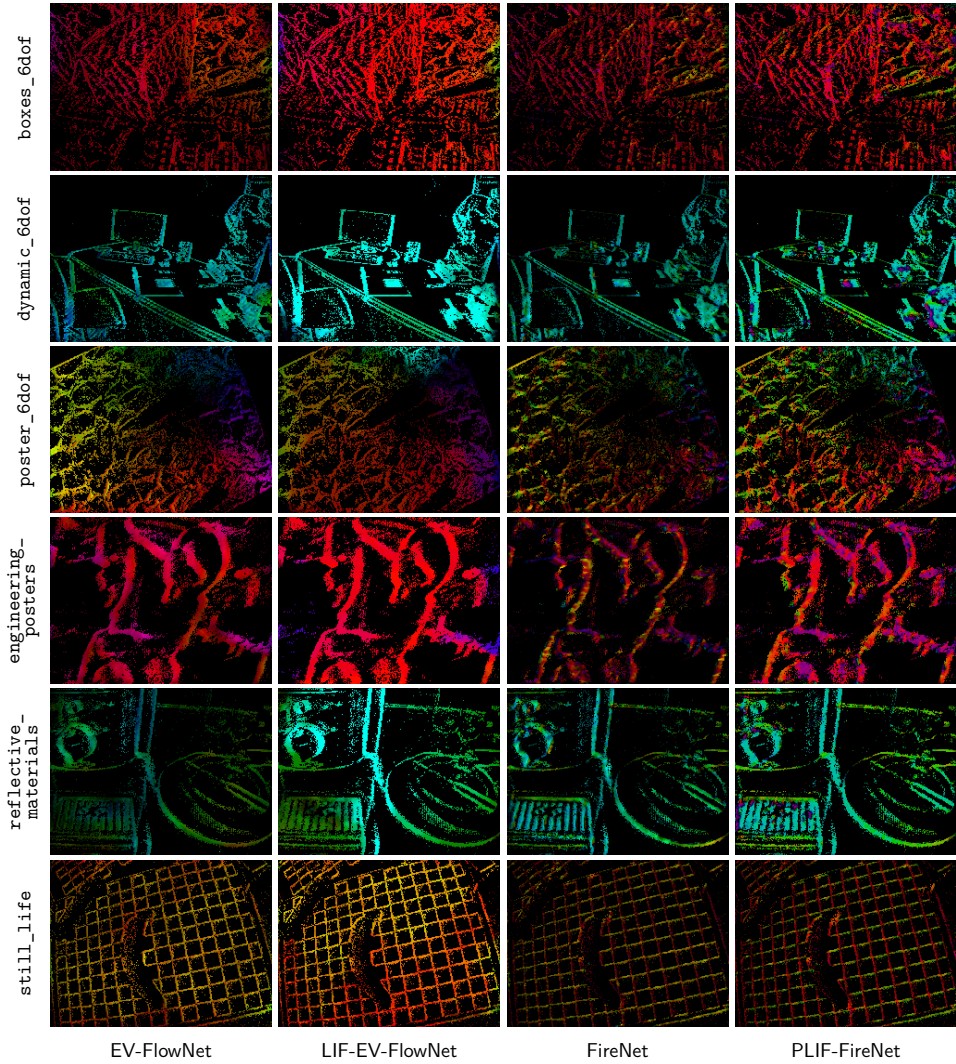

Figure 6: Additional qualitative results of our best performing ANNs and SNNs on sequences from the ECD [31] (top three) and HQF [44] (bottom three) datasets.

Apart from further confirming the generalizability of our architectures to other datasets and the on-par performance of our SNNs with respect to the recurrent ANNs (and thus to the state-of-the-art), results from this evaluation reveal the lack of robustness of the self-supervised FWL metric from Stoffregen and Scheerlinck *et al.* [44] in capturing the quality of the learned event-based optical flow. As shown in Table 2, FWL results do not correlate with the AEEs reported in Table 1. For instance, FireNet variants are characterized by higher values (thus better, according to [44]) than their computationally more powerful EV-FlowNet counterparts overall, while, according to Table 1, it should be the opposite. On the other hand, according to its correlation with the reported AEEs in Table 1, RSAT, which is based on our reformulation of the self-supervised loss function from [56], is a more reliable metric to assess the quality of event-based optical flow without ground-truth data.

## D   Ablation study on recurrent connections

In this ablation study, we evaluate the importance of explicit recurrent connections for event-based optical flow estimation with ANNs and SNNs when using our input event representation (see Section 3.1) and training settings (see Section 4). To do this, we use the base FireNet architecture and its leaky and LIF variants (as introduced Section 3.4), and compare their performance on MVSEC

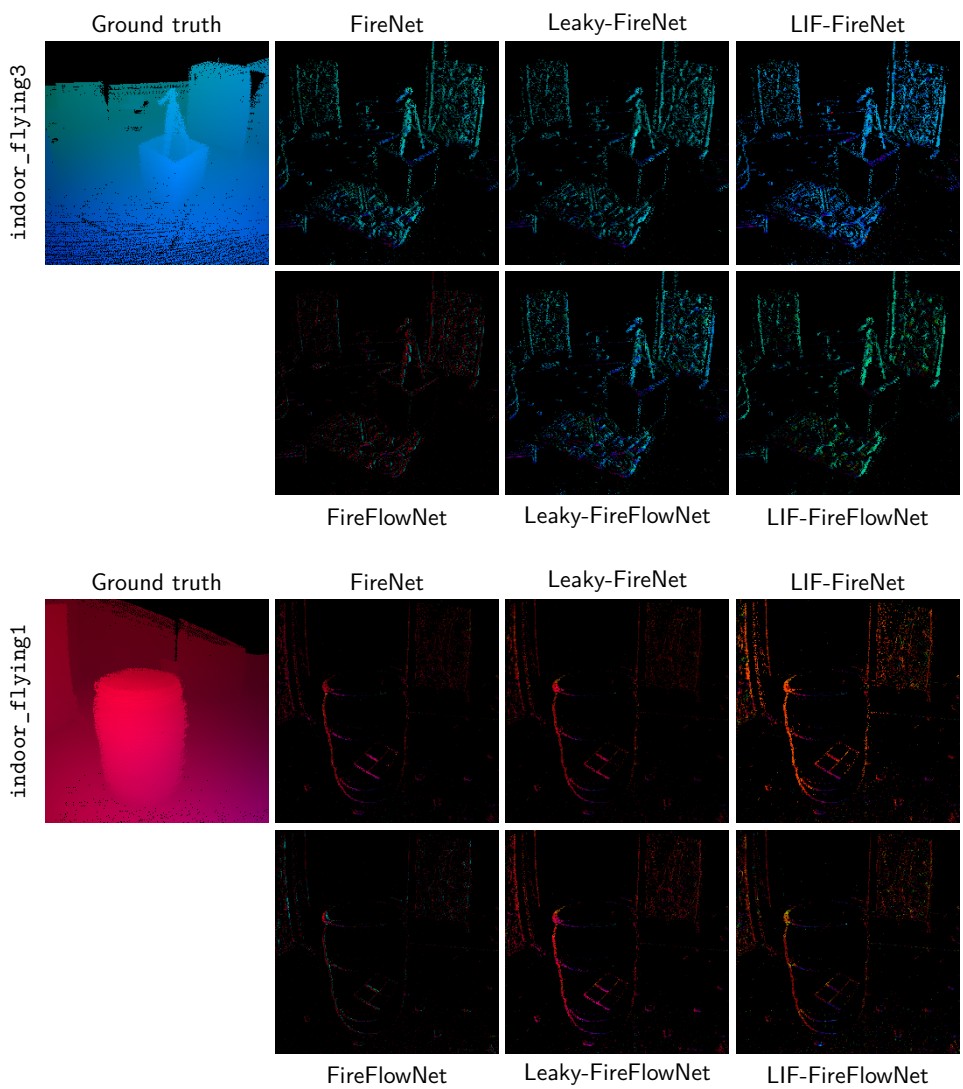

Figure 7: Qualitative results of FireNet and FireFlowNet variants on a sequence from MVSEC [55].

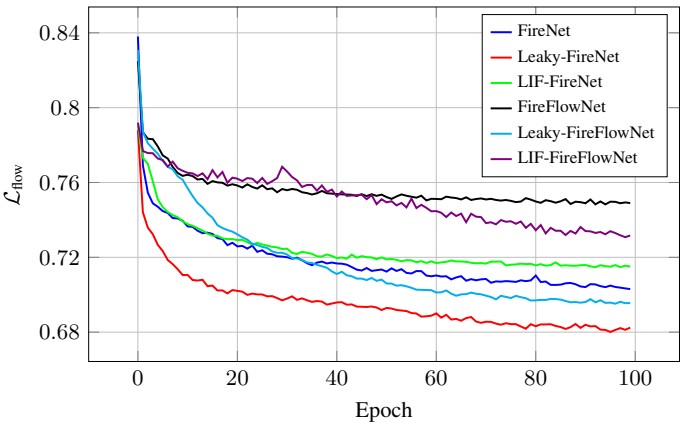

Figure 8: Training loss curves of FireNet and FireFlowNet variants.

Table 3: Quantitative evaluation of FireNet and FireFlowNet variants on MVSEC [55]. For each sequence, we report the AEE (↓) in pixels and the percentage of outliers, $\%_{\text{Outlier}}$ (↓). Best in bold, runner up underlined.

| dt = 1 | outdoor_day1 | | indoor_flying1 | | indoor_flying2 | | indoor_flying3 | |
|---|---|---|---|---|---|---|---|---|
| | AEE | $\%_{\text{Outlier}}$ | AEE | $\%_{\text{Outlier}}$ | AEE | $\%_{\text{Outlier}}$ | AEE | $\%_{\text{Outlier}}$ |
| FireNet | 0.55 | **0.35** | **0.89** | **1.93** | **1.62** | **14.65** | **1.35** | **10.64** |
| Leaky-FireNet | **0.52** | 0.41 | 0.90 | 2.66 | 1.67 | 16.09 | 1.43 | 13.16 |
| LIF-FireNet | 0.57 | 0.40 | 0.98 | 2.48 | 1.77 | 16.40 | 1.50 | 12.81 |
| FireFlowNet | 1.02 | 1.62 | 1.37 | 6.86 | 2.24 | 25.74 | 2.00 | 21.09 |
| Leaky-FireFlowNet | 0.61 | 0.56 | 0.97 | 2.71 | 1.76 | 17.68 | 1.52 | 14.16 |
| LIF-FireFlowNet | 0.84 | 1.15 | 1.22 | 5.55 | 2.06 | 22.25 | 1.80 | 18.13 |

| dt = 4 | | | | | | | | |
|---|---|---|---|---|---|---|---|---|
| FireNet | 2.04 | 20.93 | **3.35** | 42.50 | **5.71** | 61.03 | **4.68** | 53.42 |
| Leaky-FireNet | **1.96** | **18.26** | 3.42 | **42.03** | 5.92 | **58.80** | 4.98 | **52.57** |
| LIF-FireNet | 2.12 | 21.00 | 3.72 | 48.27 | 6.27 | 64.16 | 5.23 | 58.43 |
| FireFlowNet | 3.88 | 55.47 | 5.29 | 68.37 | 8.26 | 79.42 | 7.33 | 78.69 |
| Leaky-FireFlowNet | 2.29 | 24.22 | 3.68 | 47.12 | 6.29 | 62.30 | 5.37 | 58.29 |
| LIF-FireFlowNet | 3.24 | 43.08 | 4.67 | 60.34 | 7.54 | 74.68 | 6.54 | 71.45 |

[55] to their non-recurrent counterparts. As in [35], the non-recurrent version of FireNet that we use, which substitutes the ConvGRUs with convolutional encoders, is further referred to as FireFlowNet. The qualitative and quantitative results for this ablation study are shown in Fig. 7 and Table 3, respectively.

Firstly, from these results, we can conclude that stateless ANNs (such as FireFlowNet) are not capable of learning to estimate optical flow using our input event representation and training pipeline. This observation confirms the claim made in Section 3.1 about the fact that our event representation minimizes the amount of temporal information encoded in the input to the networks. Secondly, these results also confirm that, in order to successfully learn optical flow, the networks need to be able to build an internal (hidden) state through explicit recurrent connections and/or neuronal dynamics. As shown, the only architecture that is not able to learn optical flow is FireFlowNet. If this network is augmented with recurrent connections (i.e., FireNet), neuronal dynamics (i.e., Leaky-FireFlowNet, LIF-FireFlowNet), or both (i.e., Leaky-FireNet, LIF-FireNet), optical flow can be learned with our proposed pipeline and event representation. However, from the quantitative results in Table 3, we can observe that learning optical flow through neuronal dynamics without explicit recurrent connections (i.e., Leaky-FireFlowNet, LIF-FireFlowNet), although possible, is quite complex and results in networks with lower accuracy. For this reason, we conclude that recurrent connections are an important driver for learning accurate event-based optical flow with our training pipeline, and hence, we use them in ANNs and SNNs that we propose in this work.

Additionally, we plotted the training loss curves for the various architectures in Fig. 8. These do not paint the same picture as the quantitative evaluation results in Table 3: for instance, the AEEs of Leaky-FireFlowNet are worse than those of FireNet, but their training losses suggest otherwise. This could be caused by different networks focusing on different parts of the loss: without recurrent connections, decreasing $\mathcal{L}_{\text{contrast}}$ may be more difficult, whereas focusing efforts on $\mathcal{L}_{\text{smooth}}$ might still allow for decreasing the overall loss $\mathcal{L}_{\text{flow}}$. While resulting in similar training losses, the latter approach does not lead (as much) to the actual learning of optical flow.

## E  Ablation study on learnable parameters for SNNs

Several works emphasize the importance of including neuronal parameters in the optimization [14, 38, 49, 52], agreeing that including the various decays or leaks is beneficial for performance. Some also argue and show that learning thresholds adds little value [14, 38], which makes intuitive sense given that the same effect can be achieved through scaling the synaptic weights. To confirm these observations, we perform an ablation study on the learning of per-channel leaks and thresholds

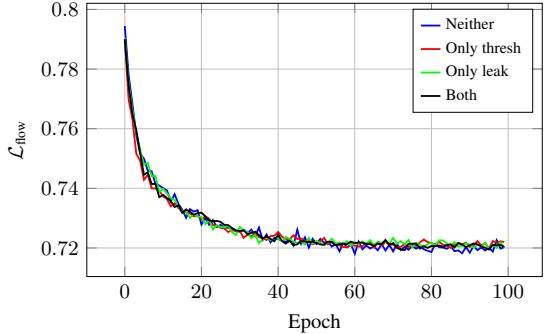

Figure 9: Training loss curves for LIF-FireNet variants with different sets of learnable parameters.

Table 4: Ablation study on learnable parameters for SNNs on MVSEC [55] using variants of LIF-FireNet. We report the AEE (↓) for dt = 1. Best in bold, runner up underlined.

| Learnable thresholds | | X | | X |
|---|---|---|---|---|
| Learnable leaks | | | X | X |
| outdoor_day1 | 0.65 | 0.68 | **0.57** | 0.58 |
| indoor_flying1 | 1.14 | 1.04 | 0.97 | **0.96** |
| indoor_flying2 | 1.88 | 1.89 | **1.70** | 1.82 |
| indoor_flying3 | 1.62 | 1.61 | **1.45** | 1.52 |

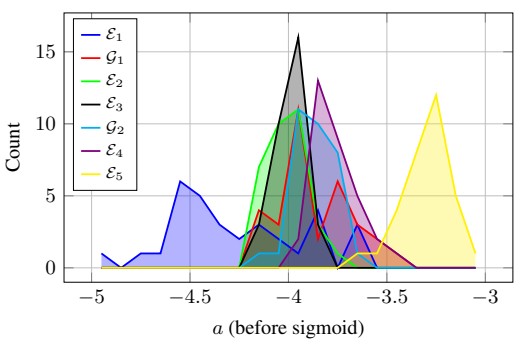

(a) Variant with learnable leaks.

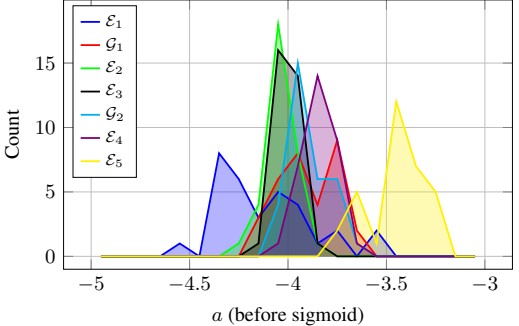

(b) Variant with learnable leaks and thresholds.

Figure 10: LIF-FireNet distribution of learned leaks $\alpha = \frac{1}{1+\exp(-a)}$, initialized at $a = -4$.

for LIF-FireNet. All instances of a parameter are initialized to the same value, but can be adapted over time in the case of learning. Initialization details can be found in Appendix G. The results in Table 4 suggest that, for our task, learning at least the leaks is beneficial for performance. However, despite these differences in AEE, the training loss curves for all variants, as shown in Fig. 9, do not vary a lot. For this we can follow the same explanation as in Appendix D: without the optimization of leaks, the network could focus on decreasing $\mathcal{L}_{\text{smooth}}$, which does not lead (as much) to the actual learning of optical flow.

Looking at the learned leaks also gives us insight into how information is integrated throughout the network. Fig. 10 shows the distribution of the parameter $a$, from which the membrane potential leaks are computed as $\alpha = \frac{1}{1+\exp(-a)}$, for the LIF-FireNet variants with learnable leaks. Initially, all $a = -4$; after learning, earlier layers mostly end up with faster leaks (lower $a$), while later layers end up with slower leaks (higher $a$). This intuitively makes sense: we want earlier layers to respond quickly to changing inputs, while we need later layers to (more slowly) integrate information over time and produce an optical flow estimate.

# F  Training loss curves of adaptive SNNs

To support the conclusions derived in Section 4.2, Fig. 11 presents the training loss curves for our EV-FlowNet and FireNet spiking architectures with the different adaptive mechanisms introduced in Section 3.3. As shown, while the curves for presynaptic adaptation (i.e., the PLIF and XLIF neuron models) are very similar to that of the LIF model, the loss curve of the ALIF model suggests the unsuitableness of postsynaptic adaptation when working with event data for optical flow estimation.

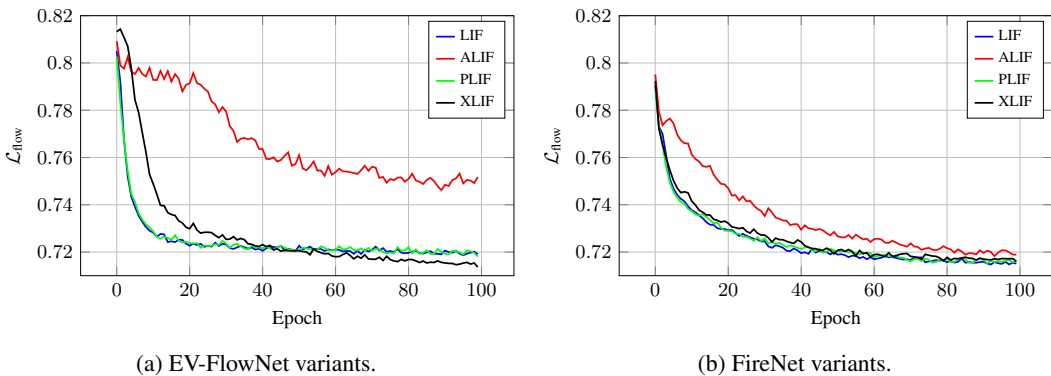

|          |          |
|:--------:|:--------:|
| (a) EV-FlowNet variants. | (b) FireNet variants. |

Figure 11: Training loss curves of our SNNs with different adaptive mechanisms.

## G    Hyperparameter and initialization details

Weights and biases of ANN modules follow the default `Conv2d` PyTorch initialization $\mathcal{U}(-1/p, 1/p)$, with $p = \sqrt{c_{in} \cdot k_1 \cdot k_2}$, $c_{in}$ the number of input channels and $\{k_1, k_2\}$ the kernel sizes. For the SNN modules, we take $p = \sqrt{c_{in}}$ for the weights to ensure enough activity, and include no biases. In the case of SNNs, we also changed the weight initialization of the prediction layer $\mathcal{P}$ to $\mathcal{U}(-0.01, 0.01)$ in order to improve learning stability, as explained in Appendix H. The aTan$'$ surrogate gradient has width $\gamma = 10$; also see Appendix H for a visualization.

Table 5 gives the leak and threshold parameters of the SNNs for the performed experiments. Membrane leak $\alpha$, threshold leak $\eta$, and trace addition/leak $\rho_{\{0,1\}}$ are clamped through a sigmoid function, e.g., $\alpha = \frac{1}{1+\exp(-a)}$, to prevent instability [14]. For $\alpha$, $\eta$ and $\rho_{\{0,1\}}$ the (learnable) parameters in the sigmoid function are $a$, $n$ and $p_{\{0,1\}}$, respectively. Threshold $\theta$ and the parameters of the adaptive threshold $\beta_{\{0,1\}}$ are clamped to $[0.01, \rightarrow)$, $[0.01, \rightarrow)$ and $[0, \rightarrow)$, respectively.

Table 5: SNN parameter initializations.

|            | Section / Appendix | |
|:----------:|:------------------:|:---:|
|            | 4.1-4.3, C, D and H | E |
| $a$        | $\mathcal{N}(-4, 0.1)$ | $-4$ |
| $n$        | $\mathcal{N}(-2, 0.1)$ | - |
| $p_0$      | $\mathcal{N}(-2, 0.1)$ | - |
| $p_1$      | $\mathcal{N}(-2, 0.1)$ | - |
| $\theta$   | $\mathcal{N}(0.8, 0.1)$ | 0.8 |
| $\beta_0$  | $\mathcal{N}(0.3, 0.1)$ | - |
| $\beta_1$  | $\mathcal{N}(1, 0.1)$ | - |

## H    Details on further lessons

We looked at the effect of surrogate gradient width on learning performance by trying out three variants on LIF-FireNet: aTan$'$ with $\gamma = 10$ (default), and SuperSpike [50] with $\gamma \in \{10, 100\}$; see Fig. 12b for a comparison of their shapes. The resulting loss curves are plotted in Fig. 12a. The width of aTan$'$-10 is such that there is sufficient gradient flow for learning; this is less so for SuperSpike-10, and not at all for SuperSpike-100. The plots of per-layer mean gradient magnitude in Fig. 13 confirm this: SuperSpike-10 only shows non-negligible gradient flow for the last two layers, while the mean gradients for SuperSpike-100 are practically zero.

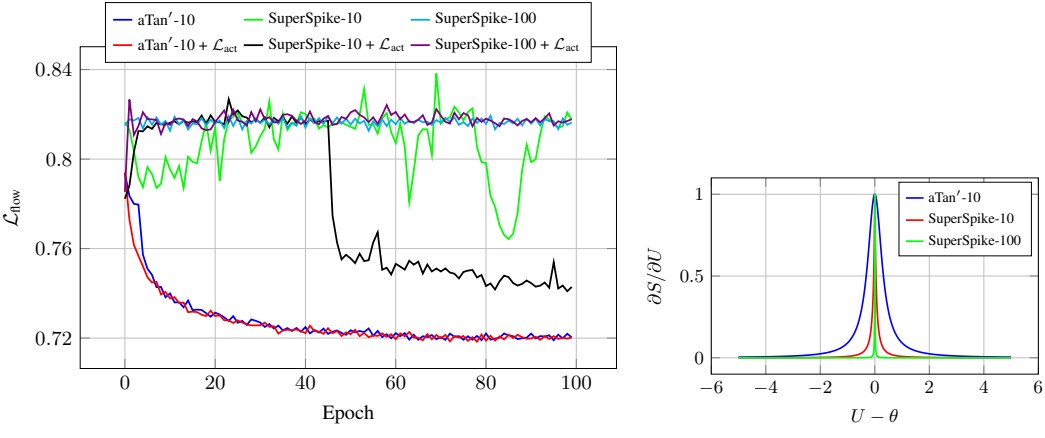

(a) Training loss curves belonging to LIF-FireNet.

(b) Surrogate gradients.

Figure 12: Impact of the choice of surrogate gradient $\sigma'$ and activity regularization $\mathcal{L}_{\text{act}}$. aTan$'$-10 denotes $\sigma'(x) = 1/(1 + 10x^2)$, where $x = U - \theta$; SuperSpike-$\gamma$ [50] denotes $\sigma'(x) = 1/(1 + \gamma|x|)^2$, with $\gamma \in \{10, 100\}$.

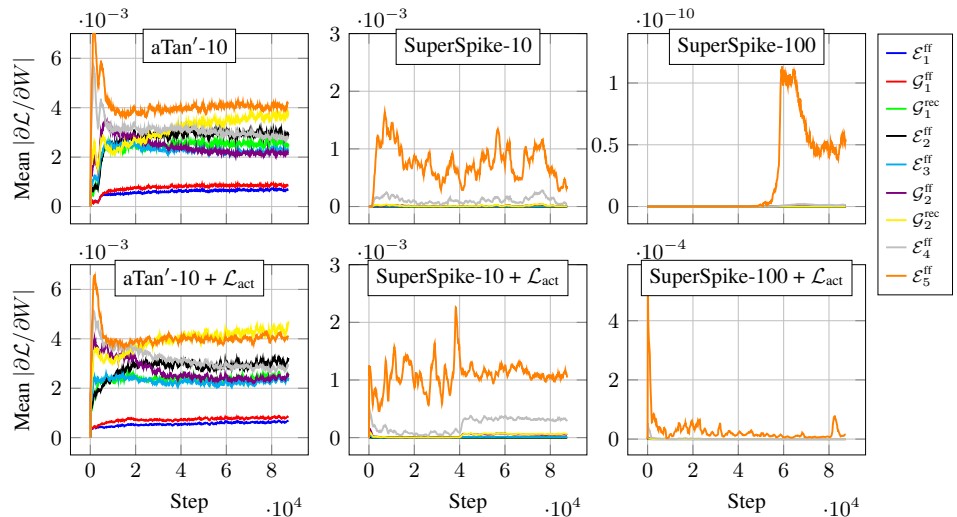

Figure 13: Per-layer mean of absolute gradient values during training of LIF-FireNet with various surrogate gradients, and activity regularization $\mathcal{L}_{\text{act}}$. aTan$'$-10 denotes $\sigma'(x) = 1/(1 + 10x^2)$, where $x = U - \theta$; SuperSpike-$\gamma$ [50] denotes $\sigma'(x) = 1/(1 + \gamma|x|)^2$, with $\gamma \in \{10, 100\}$. Data is smoothed with a 1000-step moving average and a stride of 100.

As mentioned in Section 4.3, one possible way of mitigating gradient vanishing would be to connect each layer to the loss directly, through, e.g., a regularization term on minimum activity as in [51]:

$$\mathcal{L}_{\text{act}} = \sum_{l}^{L} \max(0, f_{\text{desired}} - f_{\text{actual}})^2 \tag{17}$$

with $L$ all spiking layers, $f_{\text{desired}}$ the desired per-timestep fraction of active neurons, and $f_{\text{actual}}$ the actual per-timestep fraction of active neurons. By taking the maximum, we ensure that $\mathcal{L}_{\text{act}}$ goes to zero as soon as the activity is above the desired level. The effect of adding activity regularization with $f_{\text{desired}} = 0.05$ can be observed in Fig. 12a. While the direct connection between each layer and the loss is able to start learning for SuperSpike-10, it has little effect for SuperSpike-100. The bottom row of Fig. 13 shows that the gradient flow for SuperSpike-10 becomes non-negligible for earlier layers after step 40,000 or so; for SuperSpike-100, the gradients have increased significantly, but are still not enough to allow learning. These results are in line with the recent SNN literature, which shows that

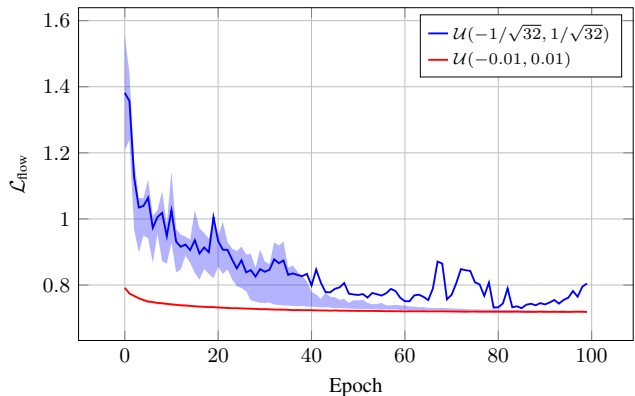

Figure 14: Mean and inter-quartile range (shaded area) of training loss curves belonging to LIF-FireNet variants with the PyTorch default weight initialization based on incoming channels and kernel size, or our initialization $\mathcal{U}(-0.01, 0.01)$, which gives much smaller weights.

SuperSpike-100 can enable learning for shallow networks [37, 51], but that it degrades performance as the number of layers increases beyond four [25], and that tuning of the surrogate width is necessary. Note that [25] also demonstrates learning with SuperSpike-10 for deeper networks, but this probably works because they use batch normalization.

Additionally, we tried different weight initializations for the prediction layer $\mathcal{P}$, based on the observation that SNNs may need smaller weights than comparable ANNs to get to similar outputs. We performed training runs with the {LIF, ALIF, PLIF, XLIF}-FireNet variants (see Table 1) and the LIF-FireNet parameter ablations (Appendix E) with (i) $\mathcal{U}(-1/\sqrt{32}, 1/\sqrt{32})$ (default PyTorch initialization), and (ii) $\mathcal{U}(-0.01, 0.01)$, which gives weights approximately 18x smaller. Fig. 14 shows the inter-quartile range (IQR) and mean for both variants.

Clearly, $\mathcal{U}(-0.01, 0.01)$ improves convergence speed and decreases variability. In fact, ALIF-FireNet with $\mathcal{U}(-1/\sqrt{32}, 1/\sqrt{32})$ failed to converge at all, hence the mean deviating from the IQR. This was not a problem with the smaller weight initialization.

# I   Comparison of activity levels for adaptive SNNs

SNNs implemented in neuromorphic hardware consume less energy as their activity decreases [11], which makes it important to investigate how activity levels vary across spiking neuron models, and how they correlate with the outputs of the network: because spiking layers emit only binary spikes, in some cases more spikes would be needed for output values larger in magnitude. We recorded the activity (fraction of nonzero values) and AEE of the {LIF, ALIF, PLIF, XLIF}-FireNet variants during the indoor_flying1 sequence of MVSEC [55] with dt = 1, as well as the mean normalized output optical flow magnitude during the boxes_6dof sequence of the ECD dataset [31] with $N = 15$k input events. Fig. 15 shows the results. One observation we can make from Fig. 15a is that the neuron models with an adaptive threshold (ALIF, XLIF) are more active than those without, while achieving similar AEEs. While this excessive spiking could be the result of initializing the base threshold $\beta_0$ too low, the similarity in AEE certainly suggests that these models spike too much for the performance they achieve, and that there is a certain redundancy in their activity.

Looking at Fig. 15b, we again observe that the models with adaptive threshold are more active than those without. On the other hand, it seems that ALIF and XLIF are more consistent in their activity across the range of outputs (narrower, more vertically oriented clusters). Looking at the clusters of LIF and PLIF, we can see that they both have roughly the same shape, but the latter's average output is larger in magnitude. This indicates that presynaptic and postsynaptic adaptive mechanisms can both serve a purpose: the former helps in increasing the absolute output range, while the latter helps in keeping activity (and therefore energy consumption) constant across this range. This makes the XLIF model especially interesting to investigate further in future work.

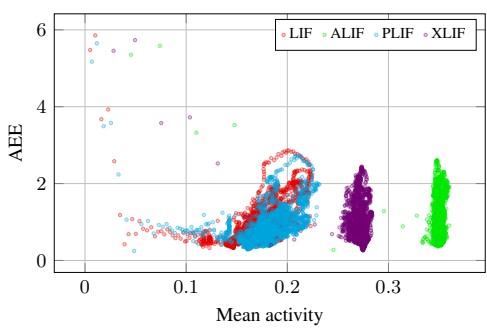
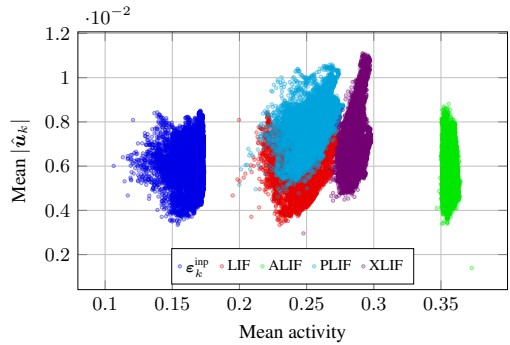

(a) AEE against mean activity (over all spiking layers).

(b) Mean (normalized) optical flow magnitude (over all pixels that have at least one event) against mean activity (over all spiking layers). The activity of the input $\varepsilon_k^{\text{inp}}$ is identical for all networks, but is here plotted against the LIF-FireNet output.

Figure 15: Recorded activity (fraction of nonzero values) of spiking FireNet variants during (a) indoor_flying1 of MVSEC [55] with dt = 1 and (b) boxes_6dof of the ECD dataset [31] with $N = 15$k events. Each marker represents one timestep.

To approximate the efficiency gains of SNNs running on neuromorphic hardware and compare it with equivalent ANN implementations, we can look at the number of accumulate (AC) and multiply-and-accumulate (MAC) operations of each, as is also done in [49]. Using energy numbers for ACs and MACs from [23], this gives us a very rough 25x increase in energy efficiency of SNNs compared to ANNs, assuming that (i) floating-point MAC operations cost five times as much energy as floating point AC operations; (ii) SNNs only make use of AC operations, while ANNs only make use of MAC operations; (iii) the average activity level of the SNN is 20%, as in Fig. 15a. However, as rightly pointed out in [11] (which contains a more elaborate quantification of efficiency gains of SNNs running on the Loihi neuromorphic processor), the comparison using AC and MAC operations for respectively SNNs and ANNs may not be a fair one for all tasks, considering, e.g., overhead in neuromorphic chips and the optimization of MACs in ANN accelerators.