# OpenReview forum: "Self-Supervised Learning of Event-Based Optical Flow with Spiking Neural Networks"
_NeurIPS.cc/2021/Conference — NeurIPS 2021 Poster_

### Official Review · Reviewer_JxQc · 2021-07-12

**Rating:** 6
**Confidence:** 2

**Summary:**

The paper investigates optical-flow estimation from event-cameras, and in particular provides the first deep SNN solution for real-world applications of this concept. The approach filters event streams from the input into an event-representation where every partition has the same number of events, which is then fed to the SNN. A backward pass is also possible using a contrast maximization loss. A new self-supervised learning framework is introduced.  The results show the new method to be on par with the state-of-the-art.

**Limitations And Societal Impact:**

The authors address limitations in their experimental evaluation (section 4.1)

**Main Review:**

The authors present the idea behind their approach, as well as the implementation details and experimental evaluation in great detail. As someone who is not fully into the related work on event-based optical flow, I found the argumentation hard to follow, but greater experts in the domain will understand this. Overall I have the impression that this advances the field of optical flow for event-based cameras, and the descriptions are fairly complete. As a suggestion I would therefore start with a shortened, more high-level explanation of the main ideas behind the approach, before going into more details in section 3.
The paper explores various variants of the approach, e.g. different network variants.

Overall I would support accepting this paper, if other more qualified reviewers agree. I my self have a hard time seeing the additional benefits over existing event-based optical flow methods. The limitations compared to standard optical flow should be more addressed.

**Time Spent Reviewing:**

2

---

> ### Author Response · Authors · 2021-08-08
> **Responses to your comments**
>
> Thank you for your review and the amount of time put in it. Please find our response to your comments below.
>
> We agree that a high-level explanation adds to the strength of the paper and makes it more accessible, which is why we included Figure 1. However, as you rightly note, this could be extended. In the revised paper, we will refine the text of the last paragraphs of the introduction to accompany this figure more clearly.
>
> Regarding the comparison to existing event-based optical flow methods, we agree that the presented work has little to show in terms of, for example, a better accuracy when estimating optical flow. However, this is not the aim of the paper. Our aim is to demonstrate the successful training of SNNs on a complex computer vision task, and to give an account and comparison of different neuron models and other settings that could help with this. It would be great if, aside from this, we could show improvements in performance, but this is secondary in our eyes, as we think the main gains can be had from using SNNs and their improvements in latency and energy efficiency once deployed on neuromorphic hardware.
>
> We hope this addresses your concerns adequately.

---

### Official Review · Reviewer_Dkce · 2021-07-16

**Rating:** 5
**Confidence:** 3

**Summary:**

The paper transforms two models for optical flow prediction into spiking neural networks (SNNs) with event-based inputs and an adapted loss function. Many of the different SNN variants largely perform relatively on par with their rate-based counterparts.

**Limitations And Societal Impact:**

The discussion of limitations is lacking, see above.


**Main Review:**

Strengths:
* applying spiking neural networks in more challenging tasks is a timely question
* the proposed models seem to perform well on a real-world dataset

Criticisms:
* lack of demonstrated improvements: As the paper points out, the premise of neuromorphic computing is more energy- or bandwidth-efficient processing. While this paper shows successful training of SNNs on a challenging task while ~maintaining performance relative to rate-based models, it falls short of demonstrating any of these improvements. I realize actually running experiments on hardware (e.g. Intel Loihi) is challenging, but approximating variables of interest is certainly doable: for instance, the required bandwidth or energy usage of each model could be approximated based on the number of events. For this work in particular, since the focus is on training, that aspect should be particularly emphasized. Otherwise, the current framing sounds more like building more complex models for the sake of it but lacks demonstrations of the motivated benefits.
* unclear presentation of specific model proposal: The analyses as well as the text outline many different variants of SNNs (RNN, Leaky, LIF, ALIF, PLIF, XLIF) on top of the two base architectures (EV-FlowNet, FireNet). Does one of these models have desirable properties over the other? The manuscript is quite hard to follow in which of the proposed models the reader should focus on. The ablation studies are unclear to me in that regard as well where stateless ANNs ("FireFlowNet") are "not capable of learning" with a 1.02 AEE compared to LIF-FireFlowNet's 0.84 AEE.

If the authors can make the specific model proposal and its benefits more (quantitatively) clear, I am willing to increase my score.

Minor:
* the AEE metric should be described in more detail. What are the units? What does a score of 1 mean? How much worse is a score of 1 compared to a score of 0.5?
* why do the reproduced numbers differ so much from the ones reported in literature? E.g. Table 1, dt=4, line 1 lists EV-FlowNet[50] with 1.23 AEE and the reproduced EV-FlowNet with 1.69 AEE. If these kinds of deviations are expected, none of the proposed models as well as the models deemed "not capable to learn" in supplemental Table 3 can be distinguished from one another.
* using the EV-FlowNet identifier for two models (from citations 50 and 52) is confusing, the differences between models should be emphasized
* typos: line 47 is missing the "a" in front of task; in line 134 "pixels locations" should be "pixel locations"


EDIT after rebuttal: I appreciate the author's arguments and have increased my score, yet still find the presentation of claims and supporting analyses supporting (see full comment).

**Time Spent Reviewing:**

5

---

> ### Author Response · Authors · 2021-08-08
> **Responses to your comments**
>
> We thank the reviewer for giving us constructive feedback and for taking the time to thoroughly read and comment on our paper. Please find our response to your comments below.
>
> First of all, we agree with the fact that, even though we studied the activity levels of the different spiking neuron models in Appendix I, we did not demonstrate that our proposed SNN architectures are more energy efficient than their ANN (i.e., rate-based) counterparts while maintaining an on-par performance. Although the focus of the paper is on the training of these SNNs and the efficiency gain comes from their actual deployment on neuromorphic hardware (see Davies *et al*., Proceedings of the IEEE 2021 for a good overview of the efficiency gains of Loihi), we agree with the reviewer that an approximation of this gain would add to the strength of this paper. One way to do this would be to compare accumulate (AC) and multiply-and-accumulate (MAC) operations between ANNs and SNNs, together with the activity sparsity of the SNN. Yin *et al*., arXiv 2021 and Sengupta *et al*., Frontiers in Neuroscience 2021 follow this approach as well, using energy numbers for AC and MAC operations from Han *et al*., NeurIPS 2015. A rough approximation in our case would be a 25x increase in energy efficiency of SNNs over ANNs, assuming that 1) MAC operations cost five times as much energy as AC operations; 2) an SNN makes only use of AC operations, while an ANN makes only use of MAC operations; 3) the average activity level of the SNN is 20%, as in Appendix I. However, as rightly pointed out in Davies *et al*., Proceedings of the IEEE 2021, the comparison using AC and MAC operations for respectively SNNs and ANNs may not be a fair one for all tasks, considering e.g. overhead in neuromorphic chips and the optimization of MACs in ANN accelerators. Because of the large uncertainty in any effiency number that we would include, we consciously decided to leave them out at submission time. To make this clear however, we will include the abovementioned considerations in the revision of the paper, together with some references on where to find good efficiency comparisons between ANNs and SNNs.
>
> Secondly, regarding the presentation of the proposed models, we would like to emphasize that the aim of this work is, generally speaking, to demonstrate SNNs on the regression problem of estimating optical flow from the sparse and asynchronous data of an event camera, to investigate which previously proposed spiking neuron models perform better than others on this complex task, and to try to find reasons for the differences. There is not a specific architecture-model combination on which the reader should focus, nor we are making a specific model proposal that should outperform the current state-of-the-art. Instead, the paper provides an extensive comparison/ablation of combinations of architectures and neuron models to study which elements of the SNN training pipeline (e.g., explicit recurrent connections, adaptive neuronal mechanisms, learnable neuronal parameters, width of the surrogate gradient) are favorable/detrimental for performance. To prevent any future misunderstanding on this topic, we will refine the text of Sections 3.3 and 3.4 in the revised version of the paper. Regarding the ablation study on recurrent connections from Appendix D, we agree with the reviewer that the quantitative comparison in Table 3 may not be all-revealing, which is why we included qualitative results for this experiment in Figure 7 (we will add more in the revised version of the paper). In these, it can be seen that all networks that have at least some form of recurrency are capable of integrating information over time and estimating optical flow, while FireFlowNet (which has neither explicit nor implicit recurrency) is not even capable of estimating the correct direction of motion, because the input representation does not explicitly encode spatiotemporal information.
>
> AEE, or average endpoint error, is defined as the euclidean distance (in pixels) between the endpoints of the predicted and ground truth optical flow vectors, averaged over all pixels with at least one input event and valid ground truth. This metric is widely used in the computer vision literature for not only assessing the accuracy of optical flow algorithms, but also for training neural networks with ground-truth data in a pure supervised fashion (e.g., Dosovitskiy *et al*., ICCV 2015). More relevant to our case, the standard benchmark for optical flow algorithms in the event-based computer vision literature consists in comparing the estimated optical flow maps against the ground-truth from the MVSEC dataset using this metric (see Zhu *et al*., RSS 2018). Therefore, we decided to use the AEE in Table 1 to assess the performance of our models and compare them to the self-supervised state-of-the-art. This quantitative evaluation is supported by the qualitative results in Figures 3 and 6. Please note that, besides this evaluation, in Appendix C we also studied the performance of our architectures on the alternative ECD and HQF datasets via the FWL (Stoffregen and Scheerlinck *et al*., ECCV 2020) and RSAT (ours) metrics, which are derived from the contrast maximization idea from Gallego *et al*., CVPR 2019. The results obtained with RSAT in Table 2 lead to the same conclusions as the AEEs in Table 1, although they are built on different principles. On the other hand, the FWL metric does not correlate with the AEE nor with RSAT, which is in line with the recent literature (i.e., Paredes-Valles, CVPR 2021).
>
> Regarding the comparison against literature results in Table 1, please note that the networks from literature were only included to give us a ballpark estimate of reasonable AEEs, and less as direct comparison. The reason for this is explained throughout Sections 3 and 4, where we say that these are different architectures (feed-forward instead of recurrent, see Section 3.4) making use of a different event representation and a different number of input events (see Section 3.1), and trained with a different loss (see third paragraph of Section 4). As indicated by the star in Table 1, our EV-FlowNet should not be seen as a reproduction of the EV-FlowNets (Zhu *et al*., RSS 2018; Zhu *et al*., CVPR 2019), but rather as a baseline for our recurrent implementation. Knowing this, Table 1 and 3 (and the associated qualitative results in Figures 3, 6, and 7) then also paint a clearer picture on the need for recurrency when using our event representation, given that FireFlowNet's errors in Table 3 are higher across all sequences compared to the other networks.
>
> Lastly, we agree with the reviewer that using the EV-FlowNet identifier for the literature and our baseline model is confusing. Although the differences between the two models are specified in Section 3.4, we will adopt different names for our recurrent version of EV-FlowNet in the revised version of the paper. Additionally, we thank the reviewer for spotting several typos. We will also amend those in our revision.
>
> We hope that we have adequately addressed your concerns. We will modify our revision as indicated.

---

> > ### Comment · Reviewer_Dkce · 2021-08-31
> > **response**
> >
> > Thank you for your response and the explanations. I appreciate the well-laid-out 25x estimate, I think this paragraph should definitely make its way into the final version of the paper in order to establish the relevance of this work.
> >
> > It may be that I am unfamiliar with the latest developments in the SNN literature, but I still find the proposed advances in this paper unclear and somewhat confusing. Neither the introduction nor the related work section made it clear IMO where the field was before and how this paper advances it.
> > The way I understand it, you benchmark a number of model variants on a complex task whereas previous work had only tested such models on simpler tasks. I agree that this is a worthwhile goal.
> >
> > However, because e.g. Table 1 is hard to parse with respect to how the different models differ from one another, I find it difficult to make specific inferences about how to proceed with respect to model development. I would encourage you to make these analyses that you are basing your claims on easier to digest, e.g. with a figure/table that groups models (by width of surrogate gradient, adaptive mechanisms, recurrence) and clearly presents how these different configurations impact performance.
> >
> > The NeurIPS format is quite unfortunate here because you cannot upload a revision with such figures/tables which might clear up my confusions. I have increased my score and will leave it up to the area chair to take this into account when making a final decision.

---

> > > ### Author Response · Authors · 2021-09-03
> > > **Response**
> > >
> > > Thank you for your thorough review and for increasing your score. We will try to accommodate your suggestions to the best of our abilities by stating more clearly our main contributions and proposed advances, and by changing the name of the models in Table 1 and Section 4 so the conclusions derived from these results are easier to understand.

---

### Official Review · Reviewer_1JoW · 2021-07-22

**Rating:** 9
**Confidence:** 5

**Summary:**

This paper modifies the basic network architecture of two mainstream self-supervised optical flow estimation methods from events in order to create a Spiking NN version of them. The paper compares several neuron models, loss functions, and gradient computation methods for training. Performance is comparable to classic CNN methods.



**Ethical Concerns:**

none.

**Limitations And Societal Impact:**

Thoroughly presented.

**Main Review:**


A really thorough paper proposing a novel SNN for self-supervised flow computation with attention to detail in implementation and exhaustive combinations of all possible neuron models and loss functions. This paper should serve as an example par excellence for experimental papers even beyond the neuromorphic community. The paper is a delight to read.

Moreover it is one of the very few papers with a full Spiking Network solution for a computer vision problem (optical flow estimation).

A SNN does not keep as input the timestamp of events. If events are arriving to the network according to their timestamps the neuron itself can encode implicitly the time of arrival. At the output, two self-supervised contrast functions are tested with a modification of the former to become convex w.r.t the flow.
LIF neurons are used for convolutions and two mainstream methods, EVflownet and Firenet, are first converted to recurrent methods and then to their SNN counterparts.


Experiments show that both SNN versions of EVflownet and Firenet are competitive with their ANNs prototypes when tested on standard benchmarks like MVSEC.
The comparison of all possible combinations/ablations in Table 1 is amazing!
The authors observe that the adaptive LIF neuron model seems detrimental for performance.
The paper identifies correctly the vanishing of gradients as a the main challenge in SNNs and proposes several heuristics.

**Time Spent Reviewing:**

4

---

> ### Author Response · Authors · 2021-08-08
> **Thanks**
>
> Thank you for your thorough review and the amount of time put in it. We feel humbled by your praise and positive feedback.

---

### Decision · Program_Chairs · 2021-09-27

**Decision:**

Accept (Poster)

**Comment:**

Dear authors,

congratulations on your paper being accepted at Neurips. The reviewers appreciated the demonstration that SNN can achieve impressive performance on a challenging and 'dense' prediction task. Please incorporate the feedback by the reviewers into the final version of the manuscript. In addition, it will be important to make the code and results publicly available, as promised in the manuscript. Your AC